# Biologically Synthesized Silver Nanoparticles and Their Diverse Applications

**DOI:** 10.3390/nano12183126

**Published:** 2022-09-09

**Authors:** Gattu Sampath, Yih-Yuan Chen, Neelamegam Rameshkumar, Muthukalingan Krishnan, Kayalvizhi Nagarajan, Douglas J. H. Shyu

**Affiliations:** 1Department of Biological Science and Technology, National Pingtung University of Science and Technology, Pingtung 912301, Taiwan; 2Department of Zoology, School of Life Sciences, Periyar University, Salem 636011, India; 3Department of Biochemical Engineering and Biotechnology, Indian Institute of Technology, New Delhi 110016, India; 4Department of Biochemical Science and Technology, National Chiayi University, Chiayi City 600355, Taiwan; 5Amity Institute of Biotechnology, Amity University, Mumbai 410206, India; 6Central University of Tamil Nadu, Thiruvarur 610005, India

**Keywords:** AgNPs, antibacterial, anticancer, photocatalytic dye degradation applications

## Abstract

Nanotechnology has become the most effective and rapidly developing field in the area of material science, and silver nanoparticles (AgNPs) are of leading interest because of their smaller size, larger surface area, and multiple applications. The use of plant sources as reducing agents in the fabrication of silver nanoparticles is most attractive due to the cheaper and less time-consuming process for synthesis. Furthermore, the tremendous attention of AgNPs in scientific fields is due to their multiple biomedical applications such as antibacterial, anticancer, and anti-inflammatory activities, and they could be used for clean environment applications. In this review, we briefly describe the types of nanoparticle syntheses and various applications of AgNPs, including antibacterial, anticancer, and larvicidal applications and photocatalytic dye degradation. It will be helpful to the extent of a better understanding of the studies of biological synthesis of AgNPs and their multiple uses.

## 1. Introduction

Nanotechnology is a key area for modern research and development, handling the synthesis and construction of particle designs ranging 1–100 nm in size [1]. The synthesis of nanoparticles (NPs) is significant in the medical field due to their vast range of chemical, physical and biological applications. NPs are differentiated based on material shape, size, and composition [2]. Several types of synthesis are being used for NPs preparations, such as chemical, physical, and biological methods [3]. The most common approach for silver nanoparticle synthesis is the chemical reduction by reducing agents (organic and inorganic). In recent years, nanobiotechnology is paying much attention to the synthesis of metal-based NPs with nanosized and multiple properties [4]. Phyto-inspired nanocomplexes or assemblies are also gaining attention due to the variety of their applications. Chen et al. (2012) reported the biomineralization process using tomato and capsicum biomolecules by synthesizing vaterite and aragonite crystals [5]. Verma et al. (2020) explained the potential antimicrobial applications using uniform Ag-doped ZnO nanostructure synthesized by *Moringa oleifera* extract [6]. Rao et al. (2012) studied the role of phyto-inspired silica nanostructures synthesized by different surfaces by using pomegranate (*Punica granatum*) leaf extract; these fabricated silica nanostructures help in enzyme immobilization applications [7].

The NPs have been effectively discovered by biological synthesis as an alternative to chemical and physical methods due to overcoming the problem of environmentally friendly, safe, cost-effective (NPs produced by waste biomass [8]) techniques for the NPs development to use in therapeutic and biomedical applications [9]. For the green synthesis of NPs, researchers have commonly used plant material [10] and bacterial [11] and fungi [12] extracts. AgNPs have remarkable considerations due to their multiple properties such as antibacterial, antifungal [13], anti-inflammatory [11,14], antibiofilm [15], anticancer [16], and larvicidal [17]. This biosynthesis process acts as the best method for AgNPs development, particularly the use of plant-extract-based AgNPs used in drug delivery [18,19]. The green synthesis of various metal-based NPs from microorganisms, plant extracts, and animal extracts are used as alternative products for the synthesis of eco-friendly NPs, they also explained various metal nanoparticles are being used for biological synthesis such as Ag, Au, CuO, MgO, Pd, Pt, NiO, ZnO, Fe_2_O_3_ NPs [20]. Among all NPs, AgNPs have more important properties and have many biological applications such as antimicrobial, anticancer, and photocatalytic activities [21].

Plant-based medicine with biomedical properties has played a potential role throughout history. The usage of herbal medicines prominently increased in recent years which showed great interest in novel drug development [22]. Ayurveda (herbal treatment to cure diseases) is being used for many disease treatments around the globe [23]. In recent years, the usage of phytomedicine has increased due to its multiple applications, low cost, and safe [24]. In certain, AgNPs synthesized by different plant extracts containing phytomedicines have been great attention due to their multiple biological applications. AgNPs synthesized by fucoidan (sulfated polysaccharide) extracted from the brown seaweed (*Spatoglossum schroederi*) showed anti-*Trypanosoma cruzi* activity [25]. The synthesis of AgNPs from pyrogallol (polyphenolic compound), purified from *Acacia nilotica* acetone leaves extract, showed multiple biological applications, such as antibacterial activity against *Helicobacter pylori*, antioxidant, and antigastric cancer activities [26]. Vanaraj et al. (2017) reported the synthesis of AgNPs from quercetin. The quercetin compound which was purified from *Clitoria ternatea* L. methanolic petal extract was used to synthesize AgNPs that showed antibiofilm activity against *Staphylococcus aureus* [27]. In this review, we briefly discussed the types of NPs synthesis, specifically focused on AgNPs synthesis from green routes, and the applications and mechanism of AgNPs in antibacterial, anticancer, antioxidant, larvicidal, and photocatalytic activities.

## 2. Types of AgNPs Synthesis

AgNPs have been studied to be used for a broad range of applications in multiple fields such as antibacterial applications, biological sensing, and catalysis applications. The different applications mainly depend on different properties such as shape, size, and the surrounding medium, which can be modified by different synthesis methods [28]. Different types of methods have been reported for AgNPs synthesis, which could be broadly divided into three different types (Figure 1) [29], including physical (laser ablation, irradiation, evaporation, condensation, and ultrasonication), biological (plant, bacteria, and fungi), and chemical (chemical reduction, sol-gel, inert condensation method, electrochemical, co-precipitation, microwave, photochemical, and pyrolysis) [30]. The chemical synthesis of AgNPs generally involves three different chemical substances, such as reducing agents, metal precursors, and capping agents. The reduction of metal salts (Ag) occurs in two stages; (1) nucleation and (2) growth [31].

### 2.1. AgNPs Synthesis by Physical and Chemical Approaches

The physical and chemical synthesis methods play the main role in NPs synthesis. The main NPs synthesis procedures include the top-down physical approach such as automatic grinding of a large number of metals, and the bottom-up chemical approaches such as metal reduction, and electrochemical process methods [32,33]. The best known method for AgNPs fabrication is a chemical reduction process. In common, diverse amounts of reducing agents such as NaBH_4_, Na_3_C_6_H_5_O_7_, C_6_H_8_O_6_, NaBH_4_, and other reagents are used for Ag^+^ to Ag reduction, and followed by accumulation into oligomeric groups, these groups ultimately form the colloidal AgNPs [34,35,36] (Figure 2).

### 2.2. Synthesis of NPs from Biological Sources

The chemical and physical methods are greatly helpful to produce the NPs, but there are some limitations such as high cost, releasing of a toxic substance into the environment, and a highly time-consuming process for NP synthesis [37]. The environmental change and the increased atmospheric temperature have raised worldwide awareness to decrease toxic waste substances, and hence, the biological method has gained more interest in the scientific field [38].

The mechanisms of AgNPs formation had been investigated extensively [39,40]. The green synthesis of AgNPs is a cost-effective and simple method accomplished by adding silver salt with biological extracts (plant, fungi, bacteria, and algae) by acting as stabilizing or capping agents. In AgNPs synthesis, the reduction of Ag^+^ to Ag^0^ has been drawn to the presence of hydroxyl functional groups in different bioactive compounds in plant and microorganism extracts. However, there is no clear AgNPs synthesis mechanism due to the presence of different biomolecules in different plants. It was reported that the formation of colloidal stable-controlled shape and size of the nanoparticles depends on the surface of the particle and the type of stabilizing agent being used. The stabilization of nanoparticles may occur due to the (1) electrostatic repulsion of particles and (2) generation of steric repulsion by non-ionic surfactants [41]. The size of the NPs could be modified by changing the different physical and chemical parameters such as pH, temperature, and the concentration of reducing agents involved in NPs synthesis [42]. The mechanism of AgNPs formation is represented in Figure 3.

Various microbes have been involved in the synthesis of many typologies of NPs, such as Au, Mg, and others [43]. Mohanpuria et al. (2008) described that fungi can produce a higher yield of NPs in contrast to bacteria because fungi secrete higher protein content, and they straightly change into greater efficiency of NPs [44]. Mukherjee et al. (2001) reported the possible synthesis mechanism AgNPs using fungi. First Ag^+^ ions interact with the surface of fungi cells, followed by reduction with the help of fungal enzymes [45]. Jyothi et al. (2016) reported that plant-based NPs are more stable than fungi and bacteria [46]. It was reported that these NPs are quickly forming, low cost, environmentally safe, and suitable for large-scale production and a one-step method process [47]. The plant extract-based NPs synthesis is divided into (1) extracellular- (raw materials), (2) intracellular- (plant tissue using cellular enzymes), and 3) phytochemical-based methods (recovered by rupturing plant cells) [48]. Prabhu and Poulose (2012) explained that different types of metabolites are the reason for the fast reduction of Ag^+^ ions compared to the microbes in the NPs synthesis process, the phytochemicals such as carboxylic acids, flavonoids, terpenoids, quinones, aldehydes, amides, and ketones are involved in the NPs synthesis mechanism [49]. The plant extract-based nanoparticles synthesis process is simply performed by mixing plant extract with metal salt at room temperature, with many various types of NPs produced by this process [50]. The synthesis of NPs could be confirmed by altering the color due to the surface plasmon atmospheric excitation, and the plasma in the free-electron system holds an equal number of easily moving electrons (e^+^) and positive ions. In the existence of electromagnetic waves, the movement of free e^+^ is driven by an electric field to constantly oscillate. These free e+ oscillations, known as plasmons, can interrelate with visible light, the process called surface plasmon resonance (SPR) [51,52]. The plant extract-based AgNPs synthesis process is shown in Figure 4.

### 2.3. Comprehensive Analyses of Commercial Products Involving Silver Nanoparticle Synthesis

Different commercial compounds used for silver nanoparticles are listed below. The major optimal conditions for the synthesis of AgNPs are also addressed to further gain insight into the pure compounds used for AgNPs production and various optimal parameters involved in the particular synthesis method (Table 1).

Until now, there is no available commercial green silver nanoparticles product on the market. However, a few silver-based biocomposites are being used for wound-dressing applications, such as PolyMem Silver^®^ (Aspen, Fort Worth, TX, USA), Tegaderm^TM^ (3M, Mapplewood, MN, USA), and Acticoat^TM^ Antimicrobial Silver Dressings (Smith & Nephew, Watford, UK) which are permitted by the Food and Drug Administration in the United States [70]. In addition to these commercial products, significant results were reported with respect to the AgNPs synthesized from biological materials for multiple biomedical applications. The AgNPs hydrogel synthesized from stabilized guar gum/curcumin composites showed significant wound healing and antibacterial activity in Wister rats [71]. The synthesis of AgNPs from *Gardenia thailandica* leaf extract showed good wound healing activity in albino rats where the excisional wounds were created and infected with *Staphylococcus aureus* [72]. The *Arthrospira platensis* (algae) supernatant extract mediated synthesized AgNPs showed good anti-breast cancer activity in the BALB/c model [73]. Moreover, AgNPs synthesized from *Musa paradisiaca* stem extract showed potential antidiabetic activity [74]. All these studies were under evaluation at the preclinical level. Further clinical investigations are required to identify their toxicity and efficacy.

## 3. Applications of AgNPs

In the earlier period, Ag was used for the treatment of different clinical disorders such as leg ulcers, acne, and epilepsy. The Ag foil was also used for surgical wound healing. The Ag and potassium nitrate pencils were used to remove ulcer debridement [75,76,77]. The AgNPs applications are mainly divided and used for therapeutic and diagnostic uses [78]. Initial findings to every infection disorder remain to play a vital role in confirming the primary action on treatment and perhaps result in an aim proved fortuitous for medication. Lin et al. (2011) explained the AgNPs used for the detection of non-invasive cancer by using surface-enhanced Raman spectroscopy [79]. Kwan et al. (2011) reported that AgNPs most effectively work on wound healing application, and they studied wound healing with AgNPs and it showed better collagen alignment after healing when compared to control [80]. Peer et al. (2007) explained that nanomedicine is estimated to improve the diagnosing of cancer by imaging and new drug design [81].

The significance of nanobiotechnology in therapeutic medicine is due to its broad-spectrum properties, cost-effectiveness, and eco-friendliness [82]. Moreover, nano-based drugs have become a major driving force behind the current developments in the drug delivery system and antibacterial therapy due to their small size and proven efficiency [83]. Presently, metal ion-based NPs possess prominent value due to their vast range of applications, including bacteria-killing properties [84]. AgNPs expanded abundant importance due to growing applications in the field of medical handlings such as in antibacterial [85] and anticancer [86] processes, the food industry [87], and consumer product development [88]. Various biological synthesis methods and their applications are represented in Figure 5.

### 3.1. Antioxidant Properties

Antioxidants are the compounds that inhibit oxidation, and many different kinds of diseases are mainly interlinked with oxidative stress caused by free radicals, which have very little half-life time and contain damaging activity towards different molecules such as lipids, proteins, and DNA. The free radicals are derived from oxygen/nitrogen and the most general ROS include hydrogen peroxide (H_2_O_2_) and superoxide anions. The ROS distributed into the body and were enabled to react to the electrons of other molecules and affected different enzymes leading to tissue damage, which may play a significant role in different disease pathogenesis (cancer, cardiovascular, neurodegenerative, and effect aging mechanisms in a living organism) [89,90,91]. Adeshina et al. (2010) explained that plant phytochemicals, such as terpenoids and flavonoids, play a major role in the defense against free radicals [92]. A previous study reported that AgNPs synthesized from the *Acacia nilotica* plant extract–purified compound pyrogallol showed antioxidant activity of 79.75 ± 1.5% at a 90 μg/mL concentration, which proved that the polyphenols are mainly involved in the antioxidant mechanism [26].

Earlier, different reports on AgNPs synthesis and its antioxidant properties, the study of the AgNPs from aqueous shoot extract of *Aristolochia bracteolate,* showed the IC_50_ of AgNPs ranged from 54.64 to 78.00 g/mL [93]. The other study showed good antioxidant activity in the synthesis of AgNPs from *Passiflora edulisf* aqueous extract, which showed 50% of DPPH antioxidant activity at a 1185.54 μg/mL concentration [94]. Other reports, such as Chinnasamy et al. (2019), explained that AgNPs synthesized by using *Melia azedarch* aqueous leaf extract showed 42.5% DPPH radical scavenging activity at a 100 μg/mL concentration [95]. Patra et al. (2019) reported that *Pisum sativum* L. outer peel mediated water extract synthesized AgNPs showed 50.17% DPPH activity at a 100 μg/mL concentration [96].

### 3.2. Antibacterial Properties of AgNPs

The unique properties of NPs have concerned potent concern for their pharmaceutical and biomedical applications such as photo-thermal, drug delivery, and bioactivity applications [97]. However, these treatments have some disadvantages such as toxic effects on non-targeted cells, drug-resistance, and cost-effectiveness. For these reasons, researchers need to find novel approaches to treat cancer. Mortezae et al. (2019) explained that the toxicity of various metal-based NPs is different; the majority of metal NPs are toxic in both lower and high concentrations at different time exposures [98]. Antimicrobial resistance has a big problem due to the lack of effective drugs against different infectious diseases causing bacteria; however, prominent development of nanomaterial-based drugs was used against drug-resistant bacteria in the past few years [99].

Before the discovery of AgNPs, the AgNO_3_ solution was topically used as an active antimicrobial agent [100]. The synthesis of AgNPs has proven to have higher antibacterial activity than the AgNO_3_ solution [101]. Thomas et al. (2007) explained that Ag is used as a disinfecting reagent and it effectively works against bacteria by blocking the bacterial respiratory chain [102]. The possible mechanisms of action of AgNPs are, firstly, the AgNPs contact with the microbe, followed by Ag offering an exceedingly larger surface area for bacterial interaction (Figure 6). Next, the NPs become stuck with bacterial cell membrane after entering inside the bacteria [103]. Second, the AgNPs Ag^+^ ions can interrelate with sulfur-containing proteins of the bacterial membranes and perhaps inhibit the function of phosphorus-containing compounds such as DNA [104]. Third, it can attack the bacterial mitochondrial respiratory chain and lead to cell death [105]. Fourth, the oxidative stress generated by ROS in bacteria results in damage to the electron transport chain due to the greater affinity of AgNPs for the cell membrane [106]. Various plants synthesizing AgNPs with antibacterial activity were reported, which are mentioned below (Table 2).

### 3.3. Use of AgNPs against Cancer

In the recent era, nanotechnology unlocks many effective treatments for cancer. Nanomedicine has displayed very good promise and growth, drastically changing the methods of cancer treatment. The mechanisms of AgNPs effects on cancer cells were investigated. It was reported that in mammalian cells, the AgNPs interrupt cell function and affect the membrane integrity by enhancing various apoptotic genes resulting in programmed cell death [127]. A report that studied the antiangiogenic properties of AgNPs explained that AgNPs induce the apoptotic pathway by ROS generation [128]. A high level of ROS production can cause cellular damage, which leads to mitochondrial membrane dysfunction [129]. In addition, AgNPs were involved in DNA damage in cancer cells [130]. The anticancer mechanism AgNPs is represented in Figure 7. Different plant extract–based AgNPs synthesis and their anticancer applications are mentioned in Table 3.

### 3.4. AgNPs Used for Controlling Mosquito Larvae

Vector-borne diseases such as zika fever and dengue fever, triggered by *Aedes aegypti*, and malaria-infected by *Culex quinquefasciatus* mosquitoes are still gaining attention nowadays. The World Health Organization reported 219 million cases of malaria in 2010 and an estimated 660,000 deaths (WHO, 2010) [147]. It was a recognized approach to efficiently eliminate the spread of vector-borne infections by controlling the number of mosquito larvae, but a chemical insecticide shows harmful and serious effects on other living organisms and the environment. Additionally, mosquitos may evolutionarily develop resistant mechanisms against various chemical insecticides. The novel plant-mediated nanoparticles might be the alternatives to developing the safety elimination approaches [148]. Vector-borne diseases such as dengue, malaria, and chikungunya are the major problem for human health due to the uncontrolled mosquito vector effects. Mosquitos developed resistance against synthetic pesticides; hence, the plant-medicated synthesis of AgNPs was proposed as the alternative vector control agents [149]. Different researchers studied AgNPs and focused their mosquito larvicidal activity by using plant extract (Table 4).

Numerous pesticide products established by medicinal plants have more significant mosquito larvicidal properties. The preparation of mosquitocidal material by purified compounds is gaining greater importance [162]. Amala and Krishnaveni (2022) reported that AgNPs from *Azolla pinnata* were tested against the IV instar larva of *A. aegypti*, which displayed lethal concentrations of LC_50_ and LC_90_ values of 2.673 and 3.255 ppm, respectively [163]. Kumar et al. (2016) also reported that *Excoecaria agallocha* leaf-mediated AgNPs were also tested against *A. aegypti* (2–14 mg/L), which showed LC_50_ value = 4.65 mg/L for exposure of 1 day [164]. Eventually, the study showed that the AgNPs synthesized from *Hibiscus rosasinensis* against *Ades albopictus* mosquito larvae exhibited larvicidal activity with minimal lethal concentration could be due to the rupture of the larval cellular membrane, DNA damage, and cell death [165].

### 3.5. AgNPs Used for Environmental Applications

The increasing water pollution has attracted the investigator to focus on the development of various photocatalysts against the chemical pollutants released by different industries in water [166]. One type of toxic chemical was the staining dyes that cause detrimental effects on aquatic organisms [167]. The research remains important to investigate the detoxification of the staining dyes to protect the water ecosystem from pollution. Various toxic staining dye-removing approaches, such as redox treatment, carbon sorption, and electrocoagulation, were generally used [168,169,170], though there is an increasing challenge to develop a more operative and modest technique to eliminate toxic chemicals [171].

The biosynthesized NPs could be used in reducing and eradicating toxic chemicals from the environment [172]. Yaqoob et al. (2020) stated that the toxic chemicals from various companies are producing larger destruction to the ecosystem by discharging toxic pollutants into water. Though different dye-degradation techniques were reported, the established NPs were proved to endorse photocatalytic dye removal among all the methods [173]. The mechanism of AgNPs on dye degradation was studied. The AgNPs absorb the visible light leading to excitation of the surface electron to the higher energy state, and next this electron is accepted by O_2_ and OH ions to form radicals. These radicals target the particular dye molecules and induce dye degradation [174,175]. The mechanistic action of AgNPs on environmentally toxic dyes is represented in Figure 8. Various plant extracts that were used for photocatalytic dye degradation are summarized in Table 5.

## 4. Conclusions

In this review, several methods for the synthesis of AgNPs are explained. AgNPs have attained growing interest due to their small size and multiple biomedical applications, including antibacterial, anticancer, and environmental applications. We briefly discuss up-to-date AgNPs studies for their applications in several categories. This report will be helpful to all the researchers who are working in this particular field. Furthermore, it gives easy access to understand the inhibitory activity of various reported plant AgNPs against pathogens, including how the AgNPs exhibit their potential antibacterial activity and the mechanistic activity of AgNPs, which may help to better understand the AgNPs–pathogen interactions. Further, the production of green synthesized AgNPs for commercial approval for human usage is still in the preclinical stage. In the future, detailed short-term and long-term biologically synthesized AgNPs toxicity, efficacy, and biocompatibility will need to be investigated with a large number of clinical validations. Long-term studies of the effects would be required for the safe use of biologically synthesized AgNPs.

## Figures and Tables

**Figure 1 nanomaterials-12-03126-f001:**
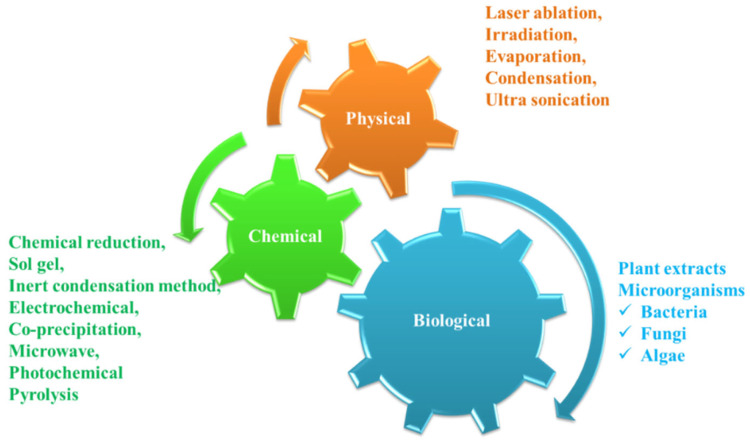
Schematic representation of biological, physical, and chemical synthetic methods of nanoparticles.

**Figure 2 nanomaterials-12-03126-f002:**
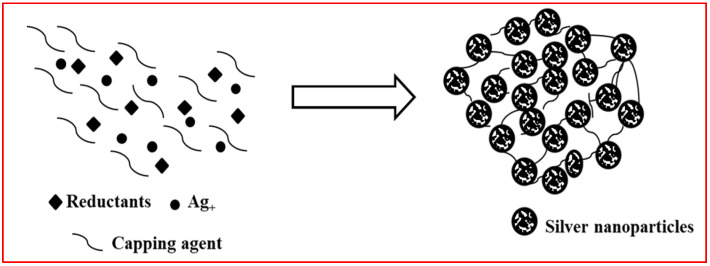
Chemical synthesis of silver nanoparticles.

**Figure 3 nanomaterials-12-03126-f003:**
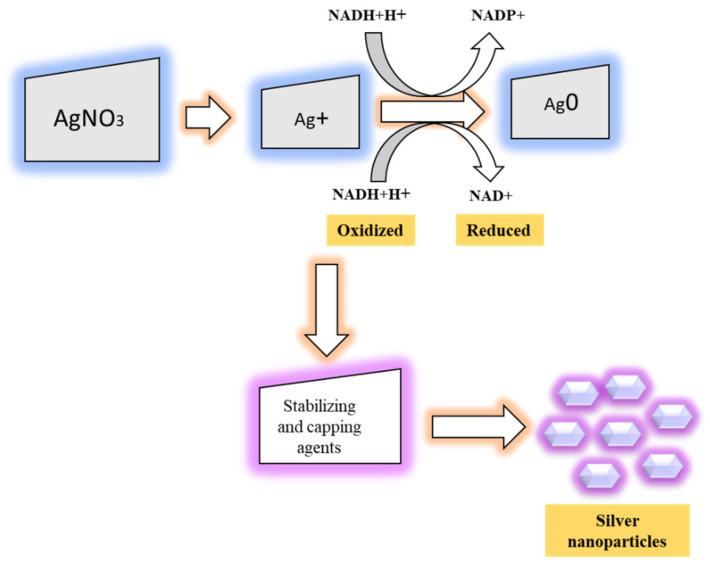
Schematic representation of mechanisms involved in AgNPs synthesis.

**Figure 4 nanomaterials-12-03126-f004:**
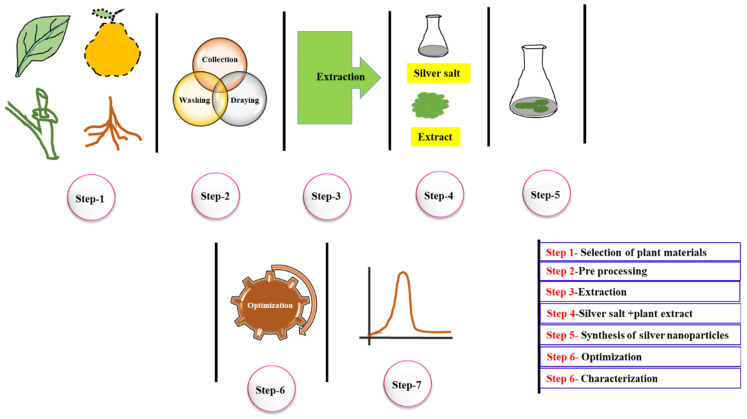
Schematic representation of plant extract-based AgNPs synthesis process.

**Figure 5 nanomaterials-12-03126-f005:**
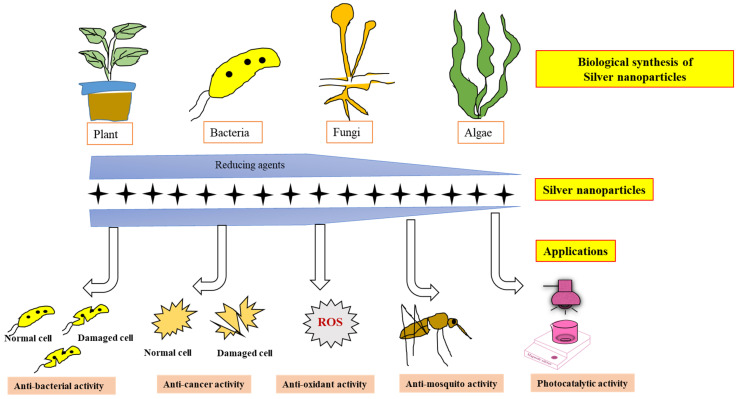
Schematic representation of types of biological AgNPs synthesis methods and their applications.

**Figure 6 nanomaterials-12-03126-f006:**
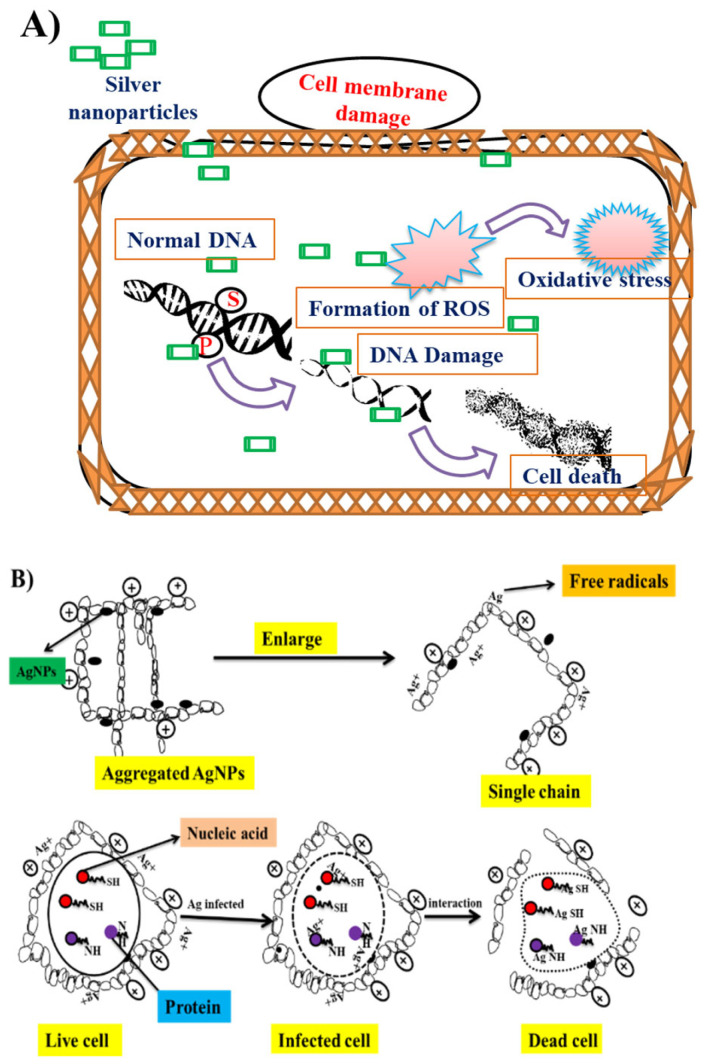
(**A**) Diagrammatic representation of the action of silver nanoparticles against bacteria; (**B**) detailed schematic representation of mechanical interactions of AgNPs with bacteria, including aggregation, generation of free radicals, infection with bacterial nucleic acids and amino groups, and bacterial cell damage.

**Figure 7 nanomaterials-12-03126-f007:**
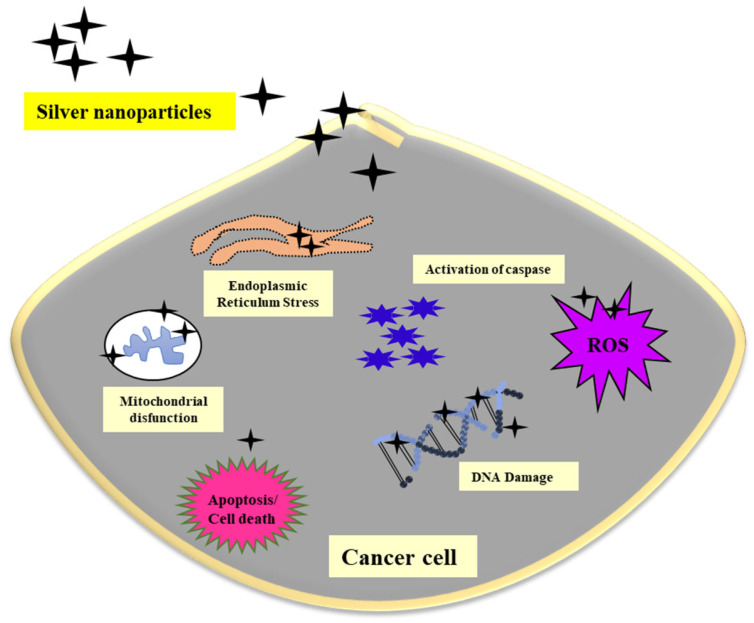
Schematic illustration of the possible mechanism of AgNPs acting on cancer cells.

**Figure 8 nanomaterials-12-03126-f008:**
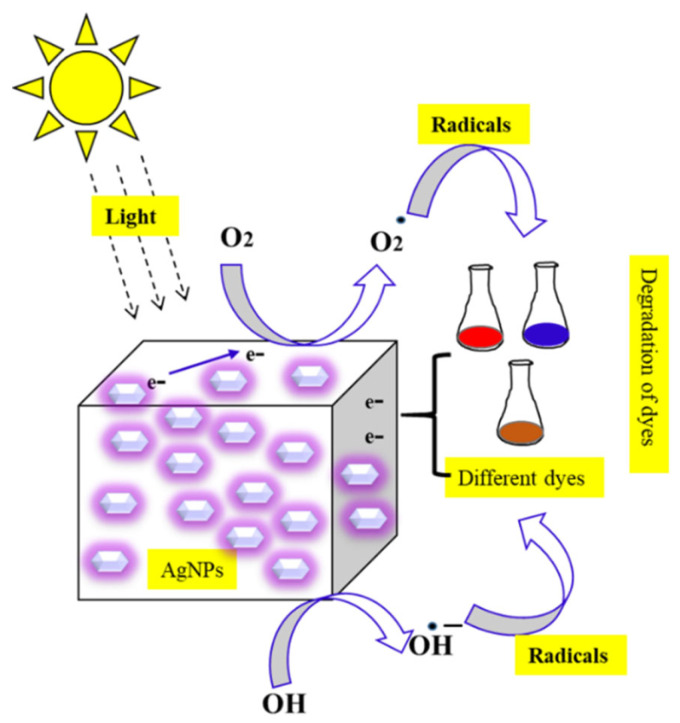
Schematic representation of the mechanism of AgNPs on photocatalytic dye degradation activity.

**Table 1 nanomaterials-12-03126-t001:** List of commercial reducing/capping agents involved in the AgNPs production.

Agents	Particle Size (nm)	Synthetic Time	Optimization Parameters Concentration/Temperature Used	Reference
Quercetin	8.4 ± 0.3	20 min	350 µM	[53]
Starch	20–60	100 min	pH = 12	[54]
Glucose	80–100	180 min	80 °C	[55]
Sucrose Maltose	6 62.4 ± 9.5	40 min 36 min	-	[56]
Ascorbic acid	31.5	15 min	6 × 10^−4^ M pH = 10.5	[57]
Curcumin	51.13	2.5 h	50 °C	[58]
Trisodium citrate	22.14	20 min	90 °C	[59]
Chitosan	5 to 30	50 min	Autoclave at a pressure of 15 psi at 120 °C	[60]
Fucoidan	39.99 ± 12.39	4 min	Microwave irradiation RT	[61]
Tannic acid	27.7–46.7	30 min	85 °C oven	[62]
Ellagic acid	-	20 min	5 to 15 µM	[63]
Rosmarinic acid	2–5	10 min	RT	[64]
Picric acid	30	4 min	-	[65]
Resveratrol	11.5 ± 3.18	2 h	80 °C oven; 0.7 µM	[66]
Sodium citrate Epigenin	95.5 93.94	12 h	RT	[67]
Bovine serum albumin	113.3	12 h	RT	[68]
Tannic acid and Sodium alginate	18.52 ± 0.07	200 W ultrasound for 10 min	1 mM	[69]

**Table 2 nanomaterials-12-03126-t002:** List of natural plant–synthesized AgNPs used against microorganisms.

Plant Name	Part Used	Antibacterial Activity *	Reference
*Ocimum santum*	Leaf	*E. coli*, *S. aureus*	[107]
*Cymbopogan citratus*	Leaf	*E. coli*, *S. aureus*, *S. typhi*, *C. albicans*	[108]
*Tribulus terrestris*	Fruit bodies	*S. pyogenes*, *P. aeruginosa*, *E. coli*, *B. subtilis*, *S. aureus*	[109]
*Santalum album*	Leaf	*E. coli*, *S. aureus*, *P. aeruginosa*, *A. chroococcum*, *B. licheniformis* 9555	[110]
*Solanum* *xanthocarpum*	Berry	*H. pylori*	[111]
*Eucalyptus chapmaniana*	Leaf	*E. coli*, *P. aeruginosa*, *K. pneumoniae*, *Proteus volgaris*, *S. aureus*, *C. albicans*	[112]
*Pomegranate*	Fruit	*B. subtilis*, *K. planticola*	[113]
*Plectranthus amboinicus*	Leaf	*E. coli*, *Penicillium* spp.	[114]
*Alternathera dentate*	Leaf	*E. coli*, *P. aeruginosa*, *K. pneumoniae*, *Enterococcus faecalis*	[115]
*Peganum harmala*	Seed	*H. pylori*	[116]
*Taraxacum officinale*	Floral	*E. faecalis*, *P. aeruginosa*	[117]
*Artemisia princeps*	Leaves	*H. pylori*	[118]
*Talinum triangulare*	Leaf	*S. aureus*, *E. coli*, *C. albicans*	[119]
*Swertia paniculata*	Aerial parts	*P. aeruginosa*, *K. pneumoniae*, *S. aureus*	[120]
*Acacia rigidula*	Stem Root	*E. coli ATCC11229* *P. auruginosa*, *B. subtilis*	[121]
*Senna alata*	Bark	*S. aureus*, *A. baumannii*, *E. coli*, *K. pneumoniae*, *P. auruginosa*, *C. albicans*	[122]
*Catharanthus roseus*	Leaf	*S. dysenteriae*, *K. pneumoniae*, *B. anthraces*, *S. aureus*, *P. aeruginosa*	[123]
*Hibiscus rosasinesis*	Leaf	*E. coli*, *S. aureus*	[124]
*Tetrapleura tetraptera*	Leaf	*S. aureus*, *E. coli*, *Salmonalla* spp.	[125]
*Perovskia abrotanoides*	Plant	*S. aureus*, *B. cereus*, *E. coli*	[126]

* Abbreviations: *S. aureus*—*Staphylococcus aureus*, *E. coli*—*Escherichia coli*, *K. pneumoniae*—*Klebsiella pneumoniae*, *B. subtilis*—*Bacillus subtilis*, *H. pylori*—*Helicobacter pylori*, *C. albicans*—*Candida albicans*, *P. aeruginosa*—*Pseudomonas aeruginosa*, *S. typhi*—*Salmonella typhi*, *S. pyogenes*—*Streptococcus pyogenes*, *E. faecalis*—*Enterococcus faecalis*, *V. cholerae*—*Vibrio cholerae*, *S. dysenteriae*—*Shigella dysenteriae*, *K. planticola*—*Klebsiella planticola*, *A. chroococcum*—*Azotobacter chroococcum*, *B. licheniformis*—*Bacillus licheniformis*.

**Table 3 nanomaterials-12-03126-t003:** List of natural plant–synthesized AgNPs used against different cancer cells.

Plant Name	Extract Used	Type of Cancer Cells *	IC_50_ Value (µg/mL)	Reference
*Phytolacca decandra*	Root ethanol	A549	80	[131]
*Ulva lactuca * (Marine Macroalgae)	Aqueous	MCF-7, HT-29, Hep-2, Vero cells	37 49 12.5 95	[132]
*Citrullus colocynthis*	Fruit-Aqueous	MCF-7 Hep-G2	22.4 17.2	[133]
*Melia dubia*	Leaf-Aqueous	MCF-7	31.2	[134]
*Cucurbita maxima* *Moringa oleifera* *Acorus calamus*	Petal Leaf Rhizome	A431	82.39 ± 31.1 83.57 ± 3.9 78.58 ± 2.7	[135]
*Saccharina japonica*	Plant-Aqueous	HeLa	-	[136]
*Azadirachta indica*	Leaf-Aqueous	A549	30	[137]
*Solanum trilobatum*	Unripe-fruit-Aqueous	MCF-7	-	[138]
*Cynodon dectylon*	Leaf-Aqueous	HepG-2	45. 6	[139]
*Syzygium aromaticum*	Cloves-Aqueous	MCF-7 HEp-2	60 50	[140]
*Indigofera tinctoria*	Leaf-Aqueous	A549	56.62 ± 0.86	[141]
*Rhynchosia suaveolens*	Leaf-Aqueous	DU-145, PC-3 SKOV3 A549	4.35 7.72 4.2 24.7	[142]
*Dodonaea viscosa*	Leaf -Methanol Acetone Acetonitrile Water	A549	14 3 80 4	[143]
*Cynara scolymus*	Leaf	MCF-7	-	[144]
*Atropa acuminate*	Leaf-Aqueous	HeLa	5.418	[145]
*Putranjiva roxburghii* wall	Seed-Aqueous	MCF-7	72.32	[146]

* Note: MCF-7—Breast cancer cell line, HT-29—Human colorectal adenocarcinoma, A549—Adenocarcinomic human alveolar basal epithelial cells, HEp2—Human liver cancer cells, Vero cells—Kidney cells, HepG2—Liver cancer cells, A431—Epidermoid carcinoma, HeLa—Cervical cancer, PC-3—Human prostate cancer cells, SKOV3—Human ovarian cancer cells, MNK45—Human gastric cancer cells.

**Table 4 nanomaterials-12-03126-t004:** Different plant-synthesized AgNPs used for mosquito larvicidal activity.

Plant Name	Type of Larvae *	LC_50_ Value	Reference
*Rhizophora mucronaota*	*Aa, Cq*	0.585, 0.891 (mg/L)	[150]
*Tinospora cordifolia*	*As, Cq*	6.43, 6.96 (mg/L)	[151]
*Mimosa pudica*	*As, Cq*	13.90, 11.73 (mg/L)	[152]
*Nelumbo nucifera*	*As, Cq*	0.69 ± 0.54, 1.10 ± 0.68 (mg/L)	[153]
*Euphorbia hirta*	*As*	16.82 ppm	[154]
*Pergularia daemia*	*Aa, As*	5.12 ± 0.31, 5.35 ± 0.34 (mg/L)	[155]
*Drypetes roxbarghii*	*Cq, As*	0.8632, 0.13 ppm	[156]
*Azadirachta Indica*	*Aa, Cq*	0.006, 0.047 (mg/L)	[157]
*Cassia roxburghii*	*As, Aa, Cq*	26.35, 28.67, 31.27 (µg/mL)	[158]
*Turbunaria ornata*	*Aa, As, Cq*	0.738, 1.134, 1.494 (µg/mL)	[159]
*Holarrhena antidysenterica*	*Aa, Cq*	5.53, 9.3 ppm	[160]
*Annona reticulata*	*Aa*	4.43 (µg/mL)	[161]

* Abbreviations: *Aa*—*Aedes aegypti*, *Cq*—*Culex quinquefasciatus*, *As*—*Anopheles stephensi*.

**Table 5 nanomaterials-12-03126-t005:** Different plant-synthesized AgNPs used for environmental applications.

Plant Extract	Extract	Type of Dye Degradation *	Time	% Dye Degradation	Reference
*Solanum nigrum*	Unripe fruit	MO	6 h	-	[176]
*Lippia citriodora*	Leaf	MB	660 min	68.7	[177]
*Moringa oleifera*	Flower	MO	52 h	97	[178]
*Lagersteoemia speciosa*	Leaves	MO	310 min	10	[179]
*Camellia japanica*	Leaf	EY dye	60 min	˃97	[180]
*Carissa carandas*	Fruit	CV	150 min	100	[181]
*Prosopis juliflora*	Bark	4-Nitrophenol	80 min	90	[182]
*Trichodwsma indicum*	Leaf	MB	210 min	82	[183]
*Angelica gigas*	Ribbed stem	EY MG	180 min	67 64	[184]
*Theobroma cacao*	Pulp	MB	180 min	98.3	[185]
*Ludwigia octovalvis*	Leaf	Alizarin red Congo red Rhodamine B MB	6 h	92.3 76 91.1 94.5	[186]
*Aspilia pluriseta*	Leaf	Congo red	30 h	50	[187]
*Ehretia laevis Roxb*	Leaves	Congo red	8 h	85	[188]

* Abbreviations: MO—Methyl orange, MB—Methylene blue, MG—Malachite green.

## Data Availability

Not applicable.

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
