# Peer review of "Biologically Synthesized Silver Nanoparticles and Their Diverse Applications"

_nanomaterials, 2022, doi:10.3390/nano12183126_

Round 1

Reviewer 1 Report

Author have reviewed the present state of the art relative to the biological synthesized Ag nanoparticles, in term of production methods and application. Although the idea behind the review is good and interesting, the execution of the paper is poor, and I think that cannot be published in the present form; the reasons for this choice are the following:

1)     the number of cited paper is very high (more or less 7 pages of references), but the references are poorly described;

2)     many times authors describe that a certain result is “higher” without specifying with respect to what;

3)     there are many paragraphs that are practically empty, with only few rows of description followed by a very large table;

4)     the paragraph relative to the NPs preparation is too short and not very detailed;

5)     the number of figures is too small for a review paper;

6)     other minor revisions are reported in the attached file.

As previously stated, I suggest to reject the paper in the present form but I want encourage the authors to properly rewrite the paper and resubmit it, because the topic is very interesting and could be a useful starting point for researchers operating in that field.

Author Response

Response to Reviewer’s Comments

Manuscript ID: nanomaterials-1840257

Title: Biologically synthesized silver nanoparticles and their diverse applications

We would like to thank the reviewers for their valuable comments. We have responded constructively in a point-by-point fashion below. Revisions are highlighted in yellow color. 

Response to Reviewer’s Comments

Manuscript ID: nanomaterials-1840257

Title: Biologically synthesized silver nanoparticles and their diverse applications

We would like to thank the reviewers for their valuable comments and have responded constructively in a point-by-point fashion below. Revisions are highlighted in yellow color.

Response to Reviewer 1 comments:

Comments and Suggestions for Authors

Authors have reviewed the present state of the art relative to the biologically synthesized Ag nanoparticles, in terms of production methods and application. Although the idea behind the review is good and interesting, the execution of the paper is poor, and I think that cannot be published in the present form; the reasons for this choice are the following:

1) The number of cited papers is very high (more or less 7 pages of references), but the references are poorly described;

Response: In accordance with the reviewer’s suggestions, we describe the references clearly in the manuscript with an explanation.

2) Many times authors describe that a certain result is “higher” without specifying with respect to what

Response: In accordance with the reviewer’s suggestions, we corrected the above statement with a clear explanation.

3) There are many paragraphs that are practically empty, with only few rows of description followed by a very large table

Response: We thank the reviewer’s valuable comments. We included clear points in paragraphs for better understanding.

4) The paragraph relative to the NPs preparation is too short and not very detailed;

Response: We thank the reviewer’s valuable suggestions. We included the detailed mechanism and preparation of AgNPs.

5) The number of figures is too small for a review paper;

Response: In accordance with the reviewer’s suggestions, we incorporated the figures in the manuscript and highlighted them in yellow color.

6) Other minor revisions are reported in the attached file.

Response: Thank you for the reviewer’s kind suggestions. We made the modifications accordingly.

Response to Reviewer 1 comments:

Comments and Suggestions for Authors

Authors have reviewed the present state of the art relative to the biologically synthesized Ag nanoparticles, in terms of production methods and application. Although the idea behind the review is good and interesting, the execution of the paper is poor, and I think that cannot be published in the present form; the reasons for this choice are the following:

1) The number of cited papers is very high (more or less 7 pages of references), but the references are poorly described;

Response: In accordance with the reviewer’s suggestions, we describe the references clearly in the manuscript with an explanation.

2) Many times authors describe that a certain result is “higher” without specifying with respect to what

Response: In accordance with the reviewer’s suggestions, we corrected the above statement with a clear explanation.

3) There are many paragraphs that are practically empty, with only few rows of description followed by a very large table

Response: We thank the reviewer’s valuable comments. We included clear points in paragraphs for better understanding.

4) The paragraph relative to the NPs preparation is too short and not very detailed;

Response: We thank the reviewer’s valuable suggestions. We included the detailed mechanism and preparation of AgNPs.

5) The number of figures is too small for a review paper;

Response: In accordance with the reviewer’s suggestions, we incorporated the figures in the manuscript and highlighted them in yellow color.

6) Other minor revisions are reported in the attached file.

Response: Thank you for the reviewer’s kind suggestions. We made the modifications accordingly.

Reviewer 2 Report

In this review manuscript, the authors cover the subject of biologically synthesized silver nanoparticles and their applications.

Overall this manuscript is well present; however I find (i) lacking mechanistic aspects of nanoparticle synthesis and also (ii) key inconsistent statements and generalization which must be addressed.

Nonetheless the topic falls within the scope of MDPI Nanomaterials and the manuscript can be considered after major revisions.

1.     In the Introduction, the authors state that ‘the physically and chemically synthesized NPs are more popular, but due to the use of the toxic chemical in preparations, these methods are highly restricted in biomedical applications’.

In my view, chemical and physical routes of synthesis are more defined chemically and are consistent. In contrast, biological synthesis e.g. with cells or extracts apply heterogeneous chemical mixtures, which can result in variations in the final product.

Moreover, to validate the authors’ statement, they must provide instances of commercial products involving biological synthesis of nanoparticles. Rather my literature survey suggest that most nanoparticles used in biomedical application come from physical or chemical synthesis.

I strongly suggest that these Introduction statements be reconsidered.

2.     Also in the Introduction ‘Patra and Beak (2014) explained that in recent years nanobiotechnology is paying much attention to the synthesis of metal-based NPs with nanosized and multiple properties’

Not only nanoparticles but also organized assemblies of nanoparticles can be derived by biological routes. Such assemblies can have interesting mechanical, optical or catalytic properties and have recently gained tremendous attention. Please also address this aspect. e.g. see  Synthesis of vaterite and aragonite crystals using biomolecules of tomato and capsicum. Russian Journal of Physical Chemistry A, 86(13), 2071-2075. "Antimicrobial potential of Ag-doped ZnO nanostructure synthesized by the green method using Moringa oleifera extract." Journal of Environmental Chemical Engineering 8.3 (2020): 103730. "Phyto-inspired silica nanowires: characterization and application in lipase immobilization." ACS Applied Materials & Interfaces 4.2 (2012): 871-877.

3.       As the authors also claim, biological synthesis is considered as ‘cost-effective’ and ‘green’.  Does this cost include the cultivation of biomass (water, land, fertilizer etc.), extraction of reducing molecules and isolating the nanoparticles from the biomass? This is a key point to consider for the field before generalization. Maybe from waste biomass, these advantages are indeed valid. I am not convinced with a blanket statement (i.e. covering biomass with other uses e.g. food and feed).  

4.     Table 1. I have strong concerns here.

Disadvantages of biological routes are overlooked: (i) seasonable changes in biomass composition. (ii) the author claim biomass to be low-cost. Waste biomass is understandably low cost. But the cost to culture e.g. bacteria include the cost of running incubators i.e. energy, water and nutrients. Please refrain from such generalizations.  

On the other hand, chemical synthesis is suggested to be ‘hazardous’. Here the citrate-based method for nanoparticle synthesis is water based and widely used but does not include any toxic chemicals.

 Overall, I believe that each route of synthesis under the 3 categories is distinct, and a general comparison emerges to be misleading.

5. Chemical synthesis e.g. chemical reduction uses interaction between organic molecules and metal ions. Similar principles govern biological routes. In this view, is biological synthesis a ‘lazy approach’ of chemical synthesis, a heterogeneous organic mixture is applied for synthesis rather than a well-defined composition?Please clarify

6. Some statements appear stand alone and appear to have no connection with the theme of nanoparticle synthesis.

e.g.  In the section 3. Applications of AgNPs, the authors state ‘Herbal medicines used for various diseases are not new to people, and it has been used for many years back. As we, know India is the greatest source of many medicinal plants and World Health Organization recognizes its uses for different kinds of diseases, phytomedicines as the most needed medicines for primary health care.’  This follows with a jump to silver nanoparticles.

Also in the Introduction: ‘ a huge number of medicinal plants including Acacia nilotica [22], Eucalyptus spp [23], Glochidion candolleanum [24], and Saraca asoca have anti-bacterial applications [25]. Apart from biological applications AgNPs have environmental and mosquito larvacidal activities [26]. Again a jump from phytomedicine to silver nanoparticles.

I strongly suggest removal of such stand -alone statements. To me, this topic of this review is not associated in anyway with the efficiency of herbal medicine. In the present version, such statements appear rather misleading and confusing.

I would recommend the author to refrain from such ‘jumps’, make specific and evidence-based statements pertaining to the review theme. e.g. which commercial products use silver nanoparticle produced with phytomedicine? Were trials performed , what were their results? What are the manufacturing practices for silver nanoparticles produced with plant biomass?

7.       I strongly recommend that the author provide an comprehensive analyses of commercial products involving silver nanoparticles and the examination of production strategies. This will be a true refection of the state of art and the actual use of chemical/physical/biological routes.In the manuscript, the authors highlight only laboratory scale reports and there is no account of any actual commercial utilization/practices.

8.    The authors overlook the role of capping agents in the plant extracts, which are essential for the colloidal stability of the nanoparticles. Key mechanistic studies have revealed this and should be covered. E.g. see "Facile synthesis of size-tunable gold nanoparticles by pomegranate (Punica granatum) leaf extract: Applications in arsenate sensing." Materials Research Bulletin 48.3 (2013): 1166-1173. "Controlled spontaneous generation of gold nanoparticles assisted by dual reducing and capping agents." Gold bulletin 44.2 (2011): 119-137.

Author Response

Response to Reviewer’s Comments

Manuscript ID: nanomaterials-1840257

Title: Biologically synthesized silver nanoparticles and their diverse applications

We would like to thank the reviewers for their valuable comments. We have responded constructively in a point-by-point fashion below. Revisions are highlighted in yellow color.

Response to Reviewer 2 comments: Comments and Suggestions for Authors

In this review manuscript, the authors cover the subject of biologically synthesized silver nanoparticles and their applications.

Overall this manuscript is well present; however, I find (i) lacking mechanistic aspects of nanoparticle synthesis and also (ii) key inconsistent statements and generalization which must be addressed.

Nonetheless the topic falls within the scope of MDPI Nanomaterials and the manuscript can be considered after major revisions

*************************************************************************

1). In the Introduction, the authors state that ‘the physically and chemically synthesized NPs are more popular, but due to the use of the toxic chemical in preparations, these methods are highly restricted in biomedical applications.

In my view, chemical and physical routes of synthesis are more defined chemically and are consistent. In contrast, biological synthesis e.g. with cells or extracts apply heterogeneous chemical mixtures, which can result in variations in the final product.

Moreover, to validate the authors’ statement, they must provide instances of commercial products involving biological synthesis of nanoparticles. Rather my literature survey suggest that most nanoparticles used in biomedical application come from physical or chemical synthesis.

I strongly suggest that these Introduction statements be reconsidered.

Response: Thank you for the reviewer’s valuable comments. To the best of our knowledge, the most common approach for silver nanoparticle synthesis is the chemical reduction by reducing agents (organic and inorganic) (Iravani et al., 2014). The final products of biologically synthesized nanoparticles also exhibited consistency if the process was operated according to the reported procedures. In addition, we incorporated different commercial products being used for AgNPs synthesis mentioned in separate tables for validation.

2). Also in the Introduction ‘Patra and Beak (2014) explained that in recent years nanobiotechnology is paying much attention to the synthesis of metal-based NPs with nanosized and multiple properties’

Not only nanoparticles but also organized assemblies of nanoparticles can be derived by biological routes. Such assemblies can have interesting mechanical, optical or catalytic properties and have recently gained tremendous attention. Please also address this aspect. e.g. see Synthesis of vaterite and aragonite crystals using biomolecules of tomato and capsicum. Russian Journal of Physical Chemistry A, 86(13), 2071-2075. "Antimicrobial potential of Ag-doped ZnO nanostructure synthesized by the green method using Moringa oleifera extract." Journal of Environmental Chemical Engineering 8.3 (2020): 103730. "Phyto-inspired silica nanowires: characterization and application in lipase immobilization." ACS Applied Materials & Interfaces 4.2 (2012): 871-877

Response: We thank the reviewer’s valuable opinion. We have incorporated the phyto-inspired complexes applications in the manuscript and all the above references were cited in their respective positions. The below-mentioned text was added to the revised manuscript.

Phyto-inspired nano complexes or assembles also had keen attention due to the verity of applications. Chen et al., (2012) reported the biomineralization process using tomato and capsicum biomolecules by synthesizing the vaterite and aragonite crystals. Verma et al., (2020) explained the potential antimicrobial applications using uniform Ag-doped ZnO nanostructure synthesized by Moringa oleifera extract. Rao et al (2012) studied the role of phyto-inspired silica nanostructures synthesized by different surfaces by using pomegranate (Punica granatum) leaf extract, these fabricated silica nanostructures help in enzyme immobilization applications.

Reference:

Chen, L.; Xu, W. H.; Zhao, Y. G.; Kang, Y.; Liu, S. H.; & Zhang, Z. Y. Synthesis of vaterite and aragonite crystals using biomolecules of tomato and capsicum. Russ. J. Phys. Chem2012, 86, (13), 2071-2075. https://doi.org/10.1134/S003602441213016X

Rao, A.; Bankar, A.; Shinde, A.; Kumar, A. R.; Gosavi, S.; & Zinjarde, S. Phyto-inspired silica nanowires: characterization and application in lipase immobilization. ACS Appl. Mater. Interfaces. 20124, (2), 871-877. https://doi.org/10.1021/am201543e

Verma, R.; Chauhan, A.; Shandilya, M.; Li, X.; Kumar, R.; & Kulshrestha, S. Antimicrobial potential of Ag-doped ZnO nanostructure synthesized by the green method using Moringa oleifera extract. J. Environ. Chem. Eng2020, 8, (3), 103730. https://doi.org/10.1016/j.jece.2020.103730

3). As the authors also claim, biological synthesis is considered as ‘cost-effective’ and ‘green’.  Does this cost include the cultivation of biomass (water, land, fertilizer etc.), extraction of reducing molecules and isolating the nanoparticles from the biomass? This is a key point to consider for the field before generalization. Maybe from waste biomass, these advantages are indeed valid. I am not convinced with a blanket statement (i.e. covering biomass with other uses e.g. food and feed). 

Response: Thank you for the reviewer’s valuable comments. We corrected the statement to make it clear.

Reference:

Zuorro, A.; Iannone, A.; Natali, S.; & Lavecchia, R. Green synthesis of silver nanoparticles using bilberry and red currant waste extracts. Processes. 2019, 7(4), 193. https://doi.org/10.3390/pr7040193

4). Table 1. I have strong concerns here.

Disadvantages of biological routes are overlooked: (i) seasonable changes in biomass composition. (ii) the author claim biomass to be low-cost. Waste biomass is understandably low cost. But the cost to culture e.g. bacteria include the cost of running incubators i.e. energy, water and nutrients. Please refrain from such generalizations. 

On the other hand, chemical synthesis is suggested to be ‘hazardous’. Here the citrate-based method for nanoparticle synthesis is water based and widely used but does not include any toxic chemicals.

Overall, I believe that each route of synthesis under the 3 categories is distinct, and a general comparison emerges to be misleading.

Response: We thank the reviewer’s valuable comments. We incorporated all the above changes in the revised manuscript and refrained the table with clear statements

Methods

Chemical

Type of synthesis

Pros

Cons

Ref

Sol gel method

Ø  High purity

Ø  Economical and potential method to produce

Ø  Time consuming process

Ø  Toxic chemical usage

Modan and Plaiasu et al., 2020

Chemical precipitation

Ø  Easy

Ø  Chemical reaction takes longer time

Ahmad et al., 2018

Photochemical

Ø  Scalable

Ø  Energy effective production

Ø  No harmful reagents used

Ø  The use of sunlight is limited due to seasonal changes

Jara et al., 2021

Huang et al., 2016

Physical

Thermal decomposition, condensation

Ø  Formation of uniform size NPs

Ø  No chemical and by products required for NPs synthesis

Ø  High energy consumption

Ø  More time taking for thermal stability

Li et al., 2018

Evaporation- condensation and Laser ablation

Ø  Synthesize larger quantities of NPs

Ø  High purity

Ø  No chemical substance used

Ø  Release toxic substance to the environment

Ø  Harmful to human health

Ø  More electricity consumption

Ø  Agglomeration

Ø  Longer duration

Lee at al., 2019

Biological

Plants

Ø  Rapid, simple process

Ø  Control the size of NPs by manipulating the reaction conditions such as pH, and concentration,

Ø  Biocompatible

Ø  Complexity in determining the reactive bioactive compounds in plants, because plant extracts contains verity of phytochemical constituents

Sampath et al., 2022; Sampath et al., 2020; Sampath et al. 2021

Bacteria

Ø  Eco-friendly

Ø  Larger particle size

Ø  Impurities in NPs

Ø  Culture contamination

Ø  Less control over NPs size

Kitching et al., 2015

Ahmed et al., 2019

Fungi

Ø  Easy scale up

Ø  Smaller size then bacterial NPs

Ø  Broad NPs size

Ø  Low reproducibility

Ø  Impurities in NPs

Kitching et al., 2015

Reference:

Li, Y.; Chang, Y.; Lian, X.; Zhou, L.; Yu, Z.; Wang, H.; & An, F. Silver nanoparticles for enhanced cancer theranostics: in vitro and in vivo perspectives. J. Biomed. Nanotech. 2018, 14(9), 1515-1542. https://doi.org/10.1166/jbn.2018.2614

Jara, N.; Milán, N. S.; Rahman, A.; Mouheb, L.; Boffito, D. C.; Jeffryes, C.; & Dahoumane, S. A. Photochemical synthesis of gold and silver nanoparticles-A review. Molecules2021, 26(15), 4585. https://doi.org/10.3390/molecules26154585

Huang, M.; Du, L.; & Feng, J. X. Photochemical synthesis of silver nanoparticles/eggshell membrane composite, its characterization and antibacterial activity. Sci Adv Mater. 2016, 8(8), 1641-1647.  https://doi.org/10.1166/sam.2016.2777

Modan, E. M.; &  Plaiasu A. G. Advantages and disadvantages of chemical methods in the elaboration of nanomaterials. The Annals of “Dunarea de Jos” University of Galati. Fascicle IX, Metallurgy and Materials Science. 2020, 43(1), 53-60. https://doi.org/10.35219/mms.2020.1.08

Ahmad, F.; Idrees, F.; & Idrees, F. Recent advancements in microwave-assisted synthesis of NiO nanostructures and their super capacitor properties: A comprehensive review. Current Nanomaterials. 20183(1), 5-17. https://doi.org/10.2174/2405461503666180305161202

Ahmad, S.; Munir, S.; Zeb, N.; Ullah, A.; Khan, B.; Ali, J.; & Ali, S. Green nanotechnology: A review on green synthesis of silver nanoparticles—An ecofriendly approach. Int. J. Nanomedicine. 2019, 14, 5087. https://doi.org/10.2147/IJN.S200254

Lee, S. H.; & Jun, B. H. Silver nanoparticles: synthesis and application for nanomedicine. Int. J. Mol. Sci. 2019, 20(4), 865. https://doi.org/10.3390/ijms20040865

Sampath, G.; Govarthanan, M.; Krishnamurthy, S.; Nagarajan, P.; Rameshkumar, N.; Krishnan, M.; & Nagarajan, K. Isolation and identification of metronidazole resistance Helicobacter pylori from gastric patients in the southeastern region of India and its advanced antibacterial treatment using biological silver oxide nanoparticles. 
Biochem. Eng. J. 2022, 108445. https://doi.org/10.1016/j.bej.2022.108445

Sampath, G.; Shyu, D. J.; Rameshkumar, N.; Krishnan, M.; Raguvaran, K.; Maheshwaran, R.; & Kayalvizhi, N. Rapid biological synthesis of silver nanoparticles and it's in vitro anti-bacterial and larvicidal activities. Adv. Sci. Eng. Med. 202012(5), 593-602. https://doi.org/10.1166/asem.2020.2561

Sampath, G.; Govarthanan, M.; Rameshkumar, N.; Krishnan, M.; Alotaibi, S. H.; & Nagarajan, K. A comparative analysis of in vivo toxicity, larvicidal and catalytic activity of synthesized silver nanoparticles. Appl. Nanosci. 2021, 1-14. https://doi.org/10.1007/s13204-021-02004-1

Kitching, M.; Ramani, M.; & Marsili, E. Fungal biosynthesis of gold nanoparticles: mechanism and scale up. Microb Biotechnol2015, 8(6), 904-917. https://doi.org/10.1111/1751-7915.12151

5). Chemical synthesis e.g. chemical reduction uses interaction between organic molecules and metal ions. Similar principles govern biological routes. In this view, is biological synthesis a ‘lazy approach’ of chemical synthesis, a heterogeneous organic mixture is applied for synthesis rather than a well-defined composition? Please clarify

Response: We thank the reviewer’s valuable comments. Both chemical and biological synthesis of nanoparticle methods are having their own individual exceptional applications and properties. However, both the methods have similar principles involved in the synthesis process but the use of reducing agent and reaction conditions are different.

In the chemical synthesis process, the involvement of toxic chemicals may damage the environment by releasing hazardous substances. The methods may require a high cost to treat those substances.

In comparison, the biological synthesis process used plant, bacteria, and algae extracts to synthesize the nanoparticles. This method is generally simple, ecofriendly, and rapid. The utilization of biomass of plants and microorganisms could be an alternative to chemical methods for the synthesis of nanoparticles in environment friendly manner.

6). Some statements appear stand alone and appear to have no connection with the theme of nanoparticle synthesis.

e.g.  In the section 3. Applications of AgNPs, the authors state ‘Herbal medicines used for various diseases are not new to people, and it has been used for many years back. As we, know India is the greatest source of many medicinal plants and World Health Organization recognizes its uses for different kinds of diseases, phytomedicines as the most needed medicines for primary health care.’  This follows with a jump to silver nanoparticles.

Also in the Introduction: ‘a huge number of medicinal plants including Acacia nilotica [22], Eucalyptus spp [23], Glochidion candolleanum [24], and Saraca asoca have anti-bacterial applications [25]. Apart from biological applications AgNPs have environmental and mosquito larvacidal activities [26]. Again a jump from phytomedicine to silver nanoparticles.

I strongly suggest removal of such stand -alone statements. To me, this topic of this review is not associated in anyway with the efficiency of herbal medicine. In the present version, such statements appear rather misleading and confusing.

I would recommend the author to refrain from such ‘jumps’, make specific and evidence-based statements pertaining to the review theme. e.g. which commercial products use silver nanoparticle produced with phytomedicine? Were trials performed, what were their results? What are the manufacturing practices for silver nanoparticles produced with plant biomass?

Response: Response: We thank the reviewer’s valuable suggestions. We corrected and modified the above statements and excluded the jumps. In addition, we have incorporated the following details in the revised manuscript.

In certain, AgNPs synthesized by different plant extracts containing phytomedicines have been great attention due to their multiple biological applications.  Souza et al., 2022 reported the AgNPs synthesized by fucoidan (Sulfated polysaccharide) extracted from the brown seaweed (Spatoglossum schroederi showed anti-Trypanosoma cruzi activity. Sampath et al., 2021 explained the synthesis of AgNPs from pyrogallol (polyphenolic compound). In this report, the pyrogallol was purified from Acacia nilotica acetone leaves extract. The synthesized AgNPs showed multiple biological applications, such as antibacterial against Helicobacter pylori, anti-oxidant, and anti-gastric cancer activities.

Vanaraj et al., 2017 reported the synthesis of AgNPs from quercetin. The quercetin compound purified from Clitoria ternatea L. methonolic petal extract showed anti-biofilm activity against Staphylococcus aureus.

Reference:

Souza, A. O.; Oliveira, J. W. D. F.; Moreno, C. J. G.; de Medeiros, M. J. C.; Fernandes-Negreiros, M. M.; Souza, F. R. M.; & Rocha, H. A. O. Silver nanoparticles containing fucoidan synthesized by green method have anti-Trypanosoma cruzi activity. Nanomaterials, 2022, 12(12), 2059. https:// doi.org/10.3390/nano12122059

Sampath, G.; Shyu, D. J.; Rameshkumar, N.; Krishnan, M.; Sivasankar, P.; & Kayalvizhi, N. Synthesis and characterization of pyrogallol capped silver nanoparticles and evaluation of their in vitro anti-bacterial, anti-cancer profile against AGS cells. J Clust Sci2021, 32(3), 549-557. https://doi.org/10.1007/s10876-020-01813-8

Vanaraj, S.; Keerthana, B.B; & Preethi, K. Biosynthesis, characterization of silver nanoparticles using quercetin from Clitoria ternatea L to enhance toxicity against bacterial biofilm. J Inorg Organomet Polym. 2017, 27, 1412–1422 https://doi.org/10.1007/s10904-017-0595-8

7). I strongly recommend that the author provide an comprehensive analyses of commercial products involving silver nanoparticles and the examination of production strategies. This will be a true reflection of the state of art and the actual use of chemical/physical/biological routes. In the manuscript, the authors highlight only laboratory scale reports and there is no account of any actual commercial utilization/practices.

Response: In accordance with the reviewer’s valuable comments, we have incorporated the comprehensive commercial compounds which are being used for AgNPs synthesis. Their optimal parameters and sizes were included in separate tables, and the corresponding references were added in to the reference section.

Table:

Agents

Particle size

Time

Optimization parameters concentration/temperature used

Reference

Quercetin

8.4±0.3

20 min

350µM

Tasca and Antiochina 2020

Starch

20-60 nm

100 min

pH =12

Pascu et al., 2021

Glucose

80-100 nm

180 min

80°C

Chen et al., 2017

Sucrose

Maltose

6 nm

62.4±9.5 nm

40 min

36 min

-

Filippo et al., 2010

Ascorbic acid

31.5 nm

15 min

6 × 10−4 M

pH =10.5

Qin et al., 2010

Curcumin

51.13 nm

2.5 hr

50°C

Karan et al., 2022

Tri sodium citrate

22.14

20 min

90°C

Yerragopu et al., 2020

Tannic acid

Sodium alginate

18.52±0.07 nm

200 W ultrasound for 10 min

1 mM

Chitosan

5 to 30 nm

50 min

Autoclave at a pressure of 15 psi at 120°C

Venkatesham e al. 2014

Fucoidan

39.99 ±12.39

4 min

Microwave irradiation

RT

Rao et al., 2020

Tannic acid

27.7-46.7 nm

30 min

85°C oven

Kim et al., 2016

Ellagic acid

-

20 min

5 to 15µM

Barnaby et al., 2011

Rosmarinic acid

2-5 nm

10 min

                     RT

Bhatt et al., 2022

Picric acid

30 nm

4 min

-

Parmar et al., 2016

Resveratrol

11.5±3.18 nm

2 hr

80°C oven

0.7 µM

Park et al., 2016

Sodium citrate

Epigenin

95.5 nm

93.94 nm

12 hr

RT

Zarei et al., 2021

Bovine serum albumin

113.3 nm

12 hr

RT

Espinosa-Cristobal et al., 2015

Reference:

Tasca, F.; & Antiochia, R. Biocide activity of green quercetin-mediated synthesized silver nanoparticles. Nanomaterials. 202010(5), 909. https://doi.org/10.3390/nano10050909

Pascu, B.; Negrea, A.; Ciopec, M.; Duteanu, N.; Negrea, P.; Nemeş, N. S.; & Micle, O. A green, simple and facile way to synthesize silver nanoparticles using soluble starch. pH studies and antimicrobial applications. Materials2021, 14(16), 4765. https://doi.org/10.3390/ma14164765

Chen, Q., Liu, G., Chen, G., Mi, T., & Tai, J. Green synthesis of silver nanoparticles with glucose for conductivity enhancement of conductive ink. BioResources2017, 12(1), 608-621.

Filippo, E.; Serra, A.; Buccolieri, A.; & Manno, D. Green synthesis of silver nanoparticles with sucrose and maltose: morphological and structural characterization. 
J. Non-Cryst. Solids. 2010, 356 (6-8), 344-350. https://doi.org/10.1016/j.jnoncrysol.2009.11.021

Qin, Y.; Ji, X.; Jing, J.; Liu, H.; Wu, H.; & Yang, W. Size control over spherical silver nanoparticles by ascorbic acid reduction. Colloids Surf. A Physicochem. Eng. 2010, 372(1-3), 172-176. https://doi.org/10.1016/j.colsurfa.2010.10.013

Karan, T.; Erenler, R.; & Bozer, B. M. Synthesis and characterization of silver nanoparticles using curcumin: cytotoxic, apoptotic, and necrotic effects on various cell lines. Zeitschrift für Naturforschung C. 2022, 77,7-8, 343-350. https://doi.org/10.1515/znc-2021-0298

Yerragopu, P. S.; Hiregoudar, S.; Nidoni, U.; Ramappa, K. T.; Sreenivas, A. G.; & Doddagoudar, S. R. Chemical synthesis of silver nanoparticles using tri-sodium citrate, stability study and their characterization. International Research Journal of Pure and Applied Chemistry, 2020, 21(3), 37-50. https://doi.org/10.9734/irjpac/2020/v21i330159

Venkatesham, M.; Ayodhya, D.; Madhusudhan, A.; Veera Babu, N.; & Veerabhadram, G. A novel green one-step synthesis of silver nanoparticles using chitosan: catalytic activity and antimicrobial studies. Appl. Nanoscience2014, 4(1), 113-119. https://doi.org/10.1007/s13204-012-0180-y

Kim, T. Y.; Cha, S. H.; Cho, S.; & Park, Y. Tannic acid-mediated green synthesis of antibacterial silver nanoparticles. Arch Pharm Res. 2016, 39(4), 465-473. https://doi.org/10.1007/s12272-016-0718-8

 Barnaby, S. N.; Samantha, M. Y.; Fath, K. R.; Tsiola, A.; Khalpari, O.; & Banerjee, I. A. Ellagic acid promoted biomimetic synthesis of shape-controlled silver nanochains. Nanotechnol2011, 22(22), 225605. https://doi.org/10.1088/0957-4484/22/22/225605

Bhatt, S.; Vyas, G.; & Paul, P. Rosmarinic acid-capped silver nanoparticles for colorimetric detection of CN–and redox-modulated surface reaction-aided detection of Cr (VI) in Water. ACS omega2022, 7(1), 1318-1328. https://doi.org/10.1021/acsomega.1c05946

Parmar, A. K.; Valand, N. N.; Solanki, K. B.; & Menon, S. K. Picric acid capped silver nanoparticles as a probe for colorimetric sensing of creatinine in human blood and cerebrospinal fluid samples. Analyst2016, 141(4), 1488-1498. https://doi.org/10.1039/C5AN02303C

Park, S.; Cha, S. H.; Cho, I.; Park, S.; Park, Y.; Cho, S; & Park, Y. Antibacterial nanocarriers of resveratrol with gold and silver nanoparticles. Mater Sci Eng C Mater Biol Appl. 2016, 58, 1160-1169. https://doi.org/10.1016/j.msec.2015.09.068. 

 Zarei, M.; Karimi, E.; Oskoueian, E.; Es-Haghi, A.; & Yazdi, M. E. T. Comparative study on the biological effects of sodium citrate-based and apigenin-based synthesized silver nanoparticles. Nutrition and cancer2021, 73(8), 1511-1519. https://doi.org/10.1080/01635581.2020.1801780

Espinosa-Cristóbal, L. F.; Martínez-Castañón, G. A.; Loyola-Rodríguez, J. P.; Niño-Martínez, N.; Ruiz, F., Zavala-Alonso, N. V.; & Reyes-López, S. Y. Bovine serum albumin and chitosan coated silver nanoparticles and its antimicrobial activity against oral and nonoral bacteria. Journal of Nanomat2015, ID 420853. https://doi.org/10.1155/2015/420853

Tian, S.; Hu, Y.; Chen, X.; Liu, C.; Xue, Y.; & Han, B. Green synthesis of silver nanoparticles using sodium alginate and tannic acid: Characterization and anti-S. aureus activity. Int. J. Biol. Macromol. 2022, 195, 515-522. https://doi.org/10.1016/j.ijbiomac.2021.12.031

Rao, S. S.; Saptami, K.; Venkatesan, J.; & Rekha, P. D. Microwave-assisted rapid synthesis of silver nanoparticles using fucoidan: Characterization with assessment of biocompatibility and antimicrobial activity. Int. J. Biol. Macromol. 2020, 163, 745-755. https://doi.org/10.1016/j.ijbiomac.2020.06.230

8). The authors overlook the role of capping agents in the plant extracts, which are essential for the colloidal stability of the nanoparticles. Key mechanistic studies have revealed this and should be covered. E.g. see "Facile synthesis of size-tunable gold nanoparticles by pomegranate (Punica granatum) leaf extract: Applications in arsenate sensing." Materials Research Bulletin 48.3 (2013): 1166-1173. "Controlled spontaneous generation of gold nanoparticles assisted by dual reducing and capping agents." Gold bulletin 44.2 (2011): 119-137.

Response: We thank the reviewer’s suggestions. We have incorporated the following key mechanistic points which are involved in the NPs synthesis and cited references in the revised manuscript.

Dumor et al., 2011 explained that the formation of colloidal stable controlled shape and size of the nanoparticles depends on surface of the particle and type of stabilizing agent being used. The stabilization of nanoparticles may occur due to the 1) electro static repulsion of particles 2) generation of steric repulsion by non-ionic surfactants. Rao et al., 2014 reported the size of the NPs could be modified by changing the different physical and chemical parameters such as pH, temperature, and concentration of reducing agents involved in NPs synthesis.

Reference:

Dumur, F.; Guerlin, A.; Dumas, E.; Bertin, D.; Gigmes, D.; & Mayer, C. R. Controlled spontaneous generation of gold nanoparticles assisted by dual reducing and capping agents. Gold Bull. 2011, 44, 119-137. https://doi.org/10.1007/s13404-011-0018-5

Rao, A.; Mahajan, K.; Bankar, A.; Srikanth, R.; Kumar, A. R.; Gosavi, S.; & Zinjarde, S. Facile synthesis of size-tunable gold nanoparticles by pomegranate (Punica granatum) leaf extract: Applications in arsenate sensing. Mater. Res. Bull. 2013, 48(3), 1166-1173. https://doi.org/10.1016/j.materresbull.2012.12.025

 *****************************************************************************

Round 2

Reviewer 2 Report

In this revised manuscript entitled ‘Biologically synthesized silver nanoparticles and their diverse applications’, the authors have addressed my previous concerns, only to some extent. Whereas I welcome the added discussions on synthesis mechanisms.

Overall, I suggest that these two majors points must be addressed:

1.       I still find lacking any instance of commercial products based on biologically produced silver nanoparticles. A mention of such products or manufacturing processes is essential to convince me that such synthesis methods are commercially viable. Perhaps this method/technology is still immature for any commercial outcome thus far.  I recommend the author to elaborate on this as future outlook (e.g. as part of the conclusion section).

If there are no commercial products/processes, please state clearly so, also recommending ways to improve this.

2.       I find Table 1 as too simplistic and maybe misleading. e.g. one drawback of chemical synthesis is ‘takes longer time’. However, the reference [33] links to a very specific type of chemical synthesis. There are certainly other chemical synthesis protocols, which are quick. Therefore, reiterating my first report, this table generalizes to an extent that becomes mis-representative of the state of art.

Another key issue is that the authors compare chemical and physical methods (e.g. sol-gel and chemical precipitation) to types of biomass (e.g. bacterial and plant). In my view, this is comparing apples and oranges.

I highly recommend either rigorous modification or deletion of Table 1.

Author Response

Response to Reviewers Comments

Manuscript ID: nanomaterials-1840257-R2

Title: Biologically synthesized silver nanoparticles and their diverse applications

We would like to thank the reviewers for their valuable comments. We have responded constructively in a point-by-point fashion below. Revisions are highlighted in yellow.

**************************************************************************

Response to Reviewer 2 comments

1). I still find lacking any instance of commercial products based on biologically produced silver nanoparticles. A mention of such products or manufacturing processes is essential to convince me that such synthesis methods are commercially viable. Perhaps this method/technology is still immature for any commercial outcome thus far.  I recommend the author to elaborate on this as future outlook (e.g. as part of the conclusion section). If there are no commercial products/processes, please state clearly so, also recommending ways to improve this.

Response: We thank the reviewer’s valuable suggestions. We explained the commercial silver products further in section 2.3 on page 5 and elaborated on the future outlook in the conclusions section on page 13.

Section 2.3 (Page 5):

Until now, there is still no available commercial green silver nanoparticles product on market. However, few silver-based biocomposites were being used for wound-dressing applications, such as PolyMemÒ Silver (Aspen), TegadermTM (3M), and ActicoatTM Antimicrobial Silver Dressings (Smith & Nephew) which were permitted by the Food and Drug Administration, United States (Burdusel et al., 2018). In addition to these commercial products, significant results were reported with respect to the AgNPs synthesized from biological materials for multiple biomedical applications. The AgNPs hydrogel synthesized from stabilized guar gum/curcumin composites showed significant wound healing and antibacterial activity in Wister rats (Bhubhanil et al., 2021). The synthesis of AgNPs from Gardenia thailandica leaf extract showed good wound healing activity in albino rats which the excisional wounds were created and infected with Staphylococcus aureus (Attallah et al., 2022). The Arthrospira platensis (algae) supernatant extract mediated synthesized AgNPs showed good anti-breast cancer activity in the BALB/c model (El-Deeb et al., 2022). Moreover, AgNPs synthesized from Musa paradisiaca stem extract showed potential anti-diabetic activity (Anbazhagan et al., 2017). All these studies were under evaluation at the preclinical level. Further clinical investigations are required to identify their toxicity and efficacy.

References:

Anbazhagan, P.; Murugan, K.; Jaganathan, A.; Sujitha, V.; Samidoss, C. M.; Jayashanthani, S; & Benelli, G. Mosquitocidal, antimalarial and antidiabetic potential of Musa paradisiaca-synthesized silver nanoparticles: in vivo and in vitro approaches. Journal of Cluster Science. 2017, 28(1), 91-107. https://doi.org/10.1007/s10876-016-1047-2

Attallah, N. G.; Elekhnawy, E.; Negm, W. A.; Hussein, I. A.; Mokhtar, F. A.; & Al-Fakhrany, O. M. In vivo and in vitro antimicrobial activity of biogenic silver nanoparticles against Staphylococcus aureus clinical isolates. Pharmaceuticals. 2022, 15(2), 194. https://doi.org/10.3390/ph15020194

Bhubhanil, S.; Talodthaisong, C.; Khongkow, M.; Namdee, K.; Wongchitrat, P.; Yingmema, W; & Kulchat, S. Enhanced wound healing properties of guar gum/curcumin-stabilized silver nanoparticle hydrogels. Scientific reports. 2021, 11(1), 1-14. https://doi.org/10.1038/s41598-021-01262-x

Burdușel, A. C.; Gherasim, O.; Grumezescu, A. M.; Mogoantă, L.; Ficai, A.; & Andronescu, E. Biomedical applications of silver nanoparticles: an up-to-date overview. Nanomaterials. 2018, 8(9), 681. https://doi.org/10.3390/nano8090681

El-Deeb, N. M.; Abo-Eleneen, M. A.; Awad, O. A.; & Abo-Shady, A. M. Arthrospira platensis-mediated green biosynthesis of silver nano-particles as breast cancer controlling agent: In vitro and in vivo safety approaches. Applied Biochemistry and Biotechnology. 2022, 194(5), 2183-2203. https://doi.org/10.1007/s12010-021-03751-1

Conclusions (page 13):

Further, the production of green synthesized AgNPs for commercial approval for human usage is still in the preclinical stage. In the future, detailed short-term and long-term biologically synthesized AgNPs toxicity, efficacy, and biocompatibility will need to be investigated with a large number of clinical validations. Long-term studies of the effects would be required for the safe use of biologically synthesized AgNPs.

2).  I find Table 1 as too simplistic and maybe misleading. e.g. one drawback of chemical synthesis is ‘takes longer time’. However, the reference [33] links to a very specific type of chemical synthesis. There are certainly other chemical synthesis protocols, which are quick. Therefore, reiterating my first report, this table generalizes to an extent that becomes mis-representative of the state of art.

Another key issue is that the authors compare chemical and physical methods (e.g. sol-gel and chemical precipitation) to types of biomass (e.g. bacterial and plant). In my view, this is comparing apples and oranges.

I highly recommend either rigorous modification or deletion of Table 1.

Response: We thank and accept the reviewer’s valuable suggestions. We deleted Table 1 from the manuscript to prevent misleading information.

**************************************************************************

Round 3

Reviewer 2 Report

After two rounds of revision, I think this manuscript can be now considered for publication in MDPI Nanomaterials. I am satisfied with the major revisions  made to improve this review.